# Probabilistic Rule Realization and Selection

**Haizi Yu**[*][†]
Department of Computer Science
University of Illinois at Urbana-Champaign
Urbana, IL 61801
haiziyu7@illinois.edu

**Tianxi Li**[*]
Department of Statistics
University of Michigan
Ann Arbor, MI 48109
tianxili@umich.edu

**Lav R. Varshney**[†]
Department of Electrical and Computer Engineering
University of Illinois at Urbana-Champaign
Urbana, IL 61801
varshney@illinois.edu

## Abstract

Abstraction and realization are bilateral processes that are key in deriving intelligence and creativity. In many domains, the two processes are approached through *rules*: high-level principles that reveal invariances within similar yet diverse examples. Under a probabilistic setting for discrete input spaces, we focus on the rule realization problem which generates input sample distributions that follow the given rules. More ambitiously, we go beyond a mechanical realization that takes whatever is given, but instead ask for proactively selecting reasonable rules to realize. This goal is demanding in practice, since the initial rule set may not always be consistent and thus intelligent compromises are needed. We formulate both rule realization and selection as two strongly connected components within a single and symmetric bi-convex problem, and derive an efficient algorithm that works at large scale. Taking music compositional rules as the main example throughout the paper, we demonstrate our model's efficiency in not only music realization (composition) but also music interpretation and understanding (analysis).

## 1 Introduction

*Abstraction* is a conceptual process by which high-level principles are derived from specific examples; *realization*, the reverse process, applies the principles to generalize [1, 2]. The two, once combined, form the art and science in developing knowledge and intelligence [3, 4]. Neural networks have recently become popular in modeling the two processes, with the belief that the neurons, as distributed data representations, are best organized hierarchically in a layered architecture [5, 6]. Probably the most relevant such examples are auto-encoders, where the cascaded encoder and decoder respectively model abstraction and realization. From a different angle that aims for interpretability, this paper first defines a high-level data representation as a partition of the raw input space, and then formalizes abstraction and realization as bi-directional probability inferences between the raw inputs and its high-level representations.

While abstraction and realization is ubiquitous among knowledge domains, this paper embodies the two as theory and composition in music, and refers to music high-level representations as *compositional rules*. Historically, theorists [7, 8] devised rules and guidelines to describe compositional

---

[*]Equal contribution.

[†]Supported in part by the IBM-Illinois Center for Cognitive Computing Systems Research (C3SR), a research collaboration as part of the IBM Cognitive Horizons Network.

regularities, resulting in music theory that serves as the formal language to speak of music style and composers' decisions. Automatic music theorists [9–11] have also been recently developed to extract probabilistic rules in an interpretable way. Both human theorists and auto-theorists enable teaching of music composition via rules such as avoiding parallel octaves and resolving tendency tones. So, writing music, to a certain extent (e.g. realizing a part-writing exercise), becomes the process of generating "legitimate" music realizations that satisfy the given rules.

This paper focuses on the realization process in music, assuming rules are given by a preceding abstraction step. There are two main challenges. First, *rule realization*: problem occurs when one asks for efficient and diverse music generation satisfying the given rules. Depending on the rule representation (hard or probabilistic), there are search-based systems that realize hard-coded rules to produce music pieces [12, 13], as well as statistical models that realize probabilistic rules to produce distributions of music pieces [9, 14]. Both types of realizations typically suffer from the enormity of the sample space, a curse of input dimensionality. Second, *rule selection* (which is subtler): not all rules are equally important nor are they always consistent. In some cases, a perfect and all-inclusive realization is not possible, which requires relaxation/sacrifice of some rules. In other cases, composers intentionally break certain rules to establish unique styles. So the freedom and creativity in selecting the "right" rules for realization poses the challenge.

The main contribution of the paper is to propose and implement a unified framework that makes reasonable rule selections and realizes them in an efficient way, tackling the two challenges in one shot. As one part of the framework, we introduce a two-step dimensionality reduction technique—a group de-overlap step followed by a screening step—to efficiently solve music rule realization. As the other part, we introduce a group-level generalization of the elastic net penalty [15] to weight the rules for a reasonable selection. The unified framework is formulated as a single bi-convex optimization problem (w.r.t. a probability variable and a weight variable) that coherently couples the two parts in a symmetric way. The symmetry is beneficial in both computation and interpretation. We run experiments on artificial rule sets to illustrate the operational characteristics of our model, and further test it on a real rule set that is exported from an automatic music theorist [11], demonstrating the model's selectivity in music rule realization at large scale.

Although music is the main case study in the paper, we formulate the problem in generality so the proposed framework is domain-agnostic and applicable anywhere there are rules (i.e. abstractions) to be understood. Detailed discussion at the end of the paper demonstrates that the framework applies directly to general real-world problems beyond music. In the discussion, we also emphasize how our algorithm is non-trivial, not just a simple combinatorial massaging of standard models. Therefore, the techniques introduced in this paper offer broader algorithmic takeaways and are worth further studying in the future.

## 2   The Formalism: Abstraction, Realization, and Rule

**Abstraction and Realization**   We restrict our attention to raw input spaces that are discrete and finite: $\mathcal{X} = \{x_1, \ldots, x_n\}$, and assume the raw data is drawn from a probability distribution $p_{\mathcal{X}}$, where the subscript refers to the sample space (*not* a random variable). We denote a high-level representation space (of $\mathcal{X}$) by a partition $\mathcal{A}$ (of $\mathcal{X}$) and its probability distribution by $p_{\mathcal{A}}$. Partitioning the raw input space gives one way of abstracting low-level details by grouping raw data into clusters and ignoring within-cluster variations. Following this line of thought, we define an *abstraction* as the process: $(\mathcal{X}, p_{\mathcal{X}}) \to (\mathcal{A}, p_{\mathcal{A}})$ for some high-level representation $\mathcal{A}$, where $p_{\mathcal{A}}$ is inferred from $p_{\mathcal{X}}$ by summing up the probability masses within each partition cluster. Conversely, we define a *realization* as the process: $(\mathcal{A}, p_{\mathcal{A}}) \to (\mathcal{X}, p_{\mathcal{X}})$, where $p_{\mathcal{X}}$ is any probability distribution that infers $p_{\mathcal{A}}$.

**Probabilistic Compositional Rule**   To put the formalism in the context of music, we first follow the convention [9] to approach a music piece as a sequence of sonorities (a generic term for chord) and view each moment in a composition as determining a sonority that fits the existing music context. If we let $\Omega$ be a finite collection of pitches specifying the discrete range of an instrument, e.g. the collection of the 88 keys on a piano, then a $k$-part sonority—$k$ simultaneously sounding pitches—is a point in $\Omega^k$. So $\mathcal{X} = \Omega^k$ is the raw input space containing all possible sonorities. Although discrete and finite, the raw input size is typically large, e.g. $|\mathcal{X}| = 88^4$ considering piano range and 4-part chorales. Therefore, theorists have invented various music parameters such as quality and inversion, to abstract specific sonorities. In this paper, we inherit the approach in [11] to formalize a high-

level representation of $\mathcal{X}$ by a feature-induced partition $\mathcal{A}$, and call the output of the corresponding abstraction $(\mathcal{A}, p_\mathcal{A})$ a *probabilistic compositional rule*.

**Probabilistic Rule System**  The interrelation between abstraction and realization $(\mathcal{X}, p_\mathcal{X}) \leftrightarrow (\mathcal{A}, p_\mathcal{A})$ can be formalized by a linear equation: $Ap = b$, where $A \in \{0, 1\}^{m \times n}$ represents a partition ($A_{ij} = 1$ if and only if $x_j$ is assigned to the $i$th cluster in the partition), and $p = p_\mathcal{X}, b = p_\mathcal{A}$ are probability distributions of the raw input space and the high-level representation space, respectively. In the sequel, we represent a rule by the pair $(A, b)$, so realizing this rule becomes solving the linear equation $Ap = b$. More interestingly, given a set of rules: $(A^{(1)}, b^{(1)}), \ldots, (A^{(K)}, b^{(K)})$, the realization of all of them involves finding a $p$ such that $A^{(r)}p = b^{(r)}$, for all $r = 1, \ldots, K$. In this case, we form a *probabilistic rule system* by stacking all rules into one single linear system:

$$A = \begin{bmatrix} A^{(1)} \\ \vdots \\ A^{(K)} \end{bmatrix} \in \{0, 1\}^{m \times n}, \quad b = \begin{bmatrix} b^{(1)} \\ \vdots \\ b^{(K)} \end{bmatrix} \in [0, 1]^m. \tag{1}$$

We call $A_{i,:}^{(r)} p = b_i^{(r)}$ a rule *component*, and $m_r = \dim(b^{(r)})$ the size (# of components) of a rule.

## 3   Unified Framework for Rule Realization and Selection

In this section, we detail a unified framework for simultaneous rule realization and selection. Recall rules themselves can be inconsistent, e.g. rules learned from different music contexts can conflict. So given an inconsistent rule system, we can only achieve $Ap \approx b$. To best realize the possibly inconsistent rule system, we solve for $p \in \Delta^n$ by minimizing the error $\|Ap - b\|_2^2 = \sum_r \|A^{(r)}p - b^{(r)}\|_2^2$, the sum of the Brier scores from every individual rule. This objective does not differentiate rules (or their components) in the rule system, which typically yields a solution that satisfies all rules approximately and achieves a small error on average. This performance, though optimal in the averaged sense, is somewhat disappointing since most often no rule is satisfied exactly (error-free). Contrarily, a human composer would typically make a clear separation: follow some rules exactly and disregard others even at the cost of a larger realization error. The decision made on rule selection usually manifests the style of a musician and is a higher level intelligence that we aim for. In this pursuit, we introduce a fine-grained set of weights $w \in \Delta^m$ to distinguish not only individual rules but also their components. The weights are estimates of relative importance, and are further leveraged for rule selection. This yields a weighted error, which is used herein to measure realization quality:

$$\mathcal{E}(p, w; A, b) = (Ap - b)^\top \mathbf{diag}(w)(Ap - b). \tag{2}$$

If we revisit the two challenges mentioned in Sec. 1, we see that under the current setting, the first challenge concerns the curse of dimensionality for $p$, while the second concerns the selectivity for $w$. We introduce two penalty terms, one each for $p$ and $w$, to tackle the two challenges, and propose the following bi-convex optimization problem as the unified framework:

$$\begin{aligned} \text{minimize} \quad & \mathcal{E}(p, w; A, b) + \lambda_p P_p(p) + \lambda_w P_w(w) \\ \text{subject to} \quad & p \in \Delta^n, w \in \Delta^m. \end{aligned} \tag{3}$$

Despite contrasting purposes, both penalty terms, $P_p(p)$ and $P_w(w)$, adopt the same high-level strategy of exploiting *group structures* in $p$ and $w$. Regarding the curse of dimensionality, we exploit the group structure of $p$ by grouping $p_j$ and $p_{j'}$ together if the $j$th and $j'$th columns of $A$ are identical, partitioning $p$'s coordinates into $K'$ groups: $g_1', \ldots, g_{K'}'$ where $K'$ is the number of distinct columns of $A$. This grouping strategy uses the fact that in a simplex-constrained linear system, we cannot determine the individual $p_j$s within each group but only their sum. We later show (Sec. 4.1) the resulting group structure of $p$ is essential in dimensionality reduction (when $K' \ll n$) and has a deeper interpretation regarding abstraction levels. Regarding the rule-level selectivity, we exploit the group structure of $w$ by grouping weights together if they are associated with the same rule, partitioning $w$'s coordinates into $K$ groups: $g_1, \ldots, g_K$ where $K$ is the number of given rules. Based on the group structures of $p$ and $w$, we introduce their corresponding group penalties as follows:

$$P_p(p) = \|p_{g_1'}\|_1^2 + \cdots + \|p_{g_{K'}'}\|_1^2, \tag{4}$$

$$P_w'(w) = \sqrt{m_1}\|w_{g_1}\|_2^1 + \cdots + \sqrt{m_K}\|w_{g_K}\|_2^1. \tag{5}$$

One can see the symmetry here: group penalty (4) on $p$ is a squared, unweighted $L_{2,1}$-norm, which is designed to secure a unique solution that favors more randomness in $p$ for the sake of diversity in sonority generation [9]; group penalty (5) on $w$ is a weighted $L_{1,2}$-norm (group lasso), which enables rule selection. However, there is a pitfall of the group lasso penalty when deployed in Problem (3): the problem has multiple global optima that are indefinite about the number of rules to pick (e.g. selecting one rule and ten consistent rules are both optimal). To give more control over the number of selections, we finalize the penalty on $w$ as the group elastic net that blends between a group lasso penalty and a ridge penalty:

$$P_w(w) = \alpha P_w'(w) + (1-\alpha)\|w\|_2^2, \quad 0 \leq \alpha \leq 1, \tag{6}$$

where $\alpha$ balances the trade-off between rule elimination (less rules) and selection (more rules).

**Model Interpretation**    Problem (3) is a bi-convex problem: fixing $p$ it is convex in $w$; fixing $w$ it is convex in $p$. The symmetry between the two optimization variables further gives us the reciprocal interpretations of the rule realization and selection problem: given $p$, the music realization, we can *analyze* its style by computing $w$; given $w$, the music style, we can *realize* it by computing $p$ and further sample from it to obtain music that matches the style. The roles of the hyperparameters $\lambda_p$ and $(\lambda_w, \alpha)$ are quite different. In setting $\lambda_p$ sufficiently small, we secure a unique solution for the rule realization part. However, for the rule selection part, what is more interesting is that adjusting $\lambda_w$ and $\alpha$ allows us to guide the overall composition towards different directions, e.g. conservative (less strictly obeyed rules) versus liberal (more loosely obeyed rules).

**Model Properties**    We state two properties of the bi-convex problem (3) as the following theorems whose proofs can be found in the supplementary material. Both theorems involve the notion of group selective weight. We say $w \in \Delta^m$ is *group selective* if for every rule in the rule set, $w$ either drops it or selects it entirely, i.e. either $w_{g_r} = 0$ or $w_{g_r} > 0$ element-wisely, for any $r = 1, \ldots, K$. For a group selective $w$, we further define $\text{supp}_{\text{g}}(w)$ to be the selected rules, i.e. $\text{supp}_{\text{g}}(w) = \{r \mid w_{g_r} > 0 \text{ element-wisely}\} \subset \{1, \ldots, K\}$.

**Theorem 1.** *Fix any $\lambda_p > 0, \alpha \in [0,1]$. Let $(p^\star(\lambda_w), w^\star(\lambda_w))$ be a solution path to problem* (3).
*(1) $w^\star(\lambda_w)$ is group selective, if $\lambda_w > 1/\alpha$.*
*(2) $\|w_{g_r}^\star(\lambda_w)\|_2 \to \sqrt{m_r}/m$ as $\lambda_w \to \infty$, for $r = 1, \ldots, K$.*

**Theorem 2.** *For $\lambda_p = 0$ and any $\lambda_w > 0, \alpha \in [0,1]$, let $(p^\star, w^\star)$ be a solution to problem* (3). *We define $\mathcal{C} \subset 2^{\{1,\ldots,K\}}$ such that any $C \in \mathcal{C}$ is a consistent (error-free) subset of the given rule set. If $\text{supp}_{\text{g}}(w^\star) \in \mathcal{C}$, then $\sum_{r \in \text{supp}_{\text{g}}(w^\star)} m_r = \max\left\{\sum_{r \in C} m_r \mid C \in \mathcal{C}\right\}$.*

Thm. 1 implies a useful range of the $\lambda_w$-solution path: if $\lambda_w$ is too large, $w^\star$ will converge to a known value that always selects all the rules; if $\lambda_w$ is too small, $w^\star$ can lose the guarantee to be group selective. This further suggests the termination criteria used later in the experiments. Thm. 2 considers rule selection in the consistent case, where the solution selects the largest number of rule components among all other consistent rule selections. Despite the condition $\lambda_p = 0$, in practice, this theorem suggests one way of using model for a small $\lambda_p$: if the primary interest is to select consistent rules, the model is guaranteed to pick as many rule components as possible (Sec. 5.1). Yet, a more interesting application is to slightly compromise consistency to achieve better selection (Sec. 5.2).

## 4   Alternating Solvers for Probability and Weight

It is natural to solve the bi-convex problem (3) by iteratively alternating the update of one optimization variable while fixing the other, yielding two alternating solvers.

### 4.1   The $p$-Solver: for Rule Realization

If we fix $w$, the optimization problem (3) boils down to:

$$
\begin{aligned}
\text{minimize} \quad & \mathcal{E}(p, w; A, b) + \lambda_p P_p(p) \\
\text{subject to} \quad & p \in \Delta^n.
\end{aligned}
\tag{7}
$$

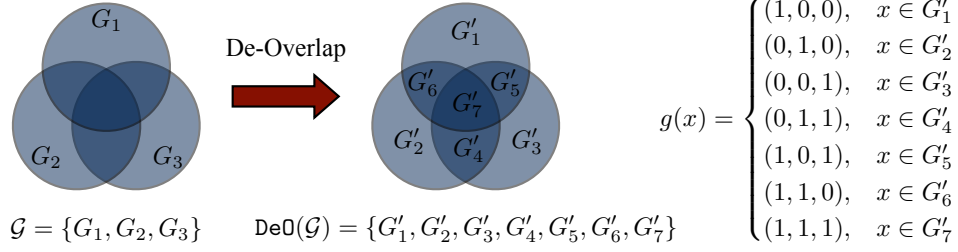

Figure 1: An example of group de-overlap.

Making a change of variable: $q_k = \mathbf{1}^\top p_{g'_k} = \|p_{g'_k}\|_1$ for $k = 1, \ldots, K'$ and letting $q = (q_1, \ldots, q_{K'})$, problem (7) is transformed to its reduced form:

$$\text{minimize} \quad \mathcal{E}(p, w; A', b) + \lambda_p \|q\|_2^2 \tag{8}$$

$$\text{subject to} \quad q \in \Delta^{K'},$$

where $A'$ is obtained from $A$ by removing its column duplicates. Problem (8) is a convex problem with a strictly convex objective, so it has a unique solution $q^\star$. However, the solution to the original problem (7) may not be unique: any $p^\star$ satisfying $q_k^\star = \mathbf{1}^\top p_{g'_k}^\star$ is a solution to (7). To favor a more random $p$ (as discussed in Sec. 3), we can uniquely determine $p^\star$ by uniformly distributing the probability mass $q_k$ within the group $g'_k$: $p_{g'_k}^\star = (q_k / \dim(p_{g'_k}))\mathbf{1}, k = 1, \ldots, K'$.

**Dimensionality Reduction: Group De-Overlap**   Problem (7) is of dimension $n$, while its reduced form (8) is of dimension $K'(\leq n)$ from which we can attain dimensionality reduction. In cases where $K' \ll n$, we have a huge speed-up for the $p$-solver; in other cases, there is still no harm to always run the $p$-solve from the reduced problem (8). Recall that we have achieved this type of dimensionality reduction by exploiting the group structure of $p$ purely from a computational perspective (Sec. 3). However, the resulting group structure has a deeper interpretation regarding abstraction levels, which is closely related to the concept of de-overlapping a family of groups, *group de-overlap* in short.

(Group De-Overlap) Let $\mathcal{G} = \{G_1, \ldots, G_m\}$ be a family of groups (a group is a non-empty set), and $G = \cup_{i=1}^m G_i$. We introduce a group assignment function $g : G \mapsto \{0, 1\}^m$, such that for any $x \in G$, $g(x)_i = \mathbb{1}\{x \in G_i\}$, and further introduce an equivalence relation $\sim$ on $G$: $x \sim x'$ if $g(x) = g(x')$. We then define the *de-overlap* of $\mathcal{G}$, another family of groups, by the quotient space

$$\text{DeO}(\mathcal{G}) = \{G'_1, \ldots, G'_{m'}\} := G/\sim . \tag{9}$$

The idea of group de-overlap is simple (Fig. 1), and $\text{DeO}(\mathcal{G})$ indeed comprises non-overlapping groups, since it is a partition of $G$ that equals the set of equivalence classes under $\sim$.

Now given a set of rules $(A^{(1)}, b^{(1)}), \ldots, (A^{(K)}, b^{(K)})$, we denote their corresponding high-level representation spaces by $\mathcal{A}^{(1)}, \ldots, \mathcal{A}^{(K)}$, each of which is a partition of the raw input space $\mathcal{X}$ (Sec. 2). Let $\mathcal{G} = \cup_{k=1}^K \mathcal{A}^{(k)}$, then $\text{DeO}(\mathcal{G})$ is a new partition—hence a new high-level representation space—of $G = \mathcal{X}$, and is finest (may be tied) among all partitions $\mathcal{A}^{(1)}, \ldots, \mathcal{A}^{(K)}$. Therefore, $\text{DeO}(\mathcal{G})$, as a summary of the rule system, delimits a lower bound on the level of abstraction produced by the given set of rules/abstractions. What coincides with $\text{DeO}(\mathcal{G})$, is the group structure of $p$ (recall: $p_j$ and $p_{j'}$ are grouped together if the $j$th and $j'$th columns of $A$ are identical), since for any $x_j \in \mathcal{X}$, the $j$th column of $A$ is precisely the group assignment vector $g(x_j)$. Therefore, the decomposed solve step from $q^\star$ to $p^\star$ reflects the following realization chain:

$$\left\{ (\mathcal{A}^{(1)}, p_{\mathcal{A}^{(1)}}), \ldots, (\mathcal{A}^{(K)}, p_{\mathcal{A}^{(K)}}) \right\} \to (\text{DeO}(\mathcal{G}), q^\star) \to (\mathcal{X}, p_{\mathcal{X}}), \tag{10}$$

where the intermediate step not only computationally achieves dimensionality reduction, but also conceptually summarizes the given set of abstractions and is further realized in the raw input space.

Note that the $\sigma$-algebra of the probability space associated with (8) is precisely generated by $\text{DeO}(\mathcal{G})$. When rules are inserted into a rule system sequentially (e.g. the growing rule set from an automatic music theorist), the successive solve of (8) is conducted along a $\sigma$-algebra path that forms a *filtration*: nested $\sigma$-algebras that lead to finer and finer delineations of the raw input space. In a pedagogical setting, the filtration reflects the iterative refinements of music composition from high-level principles that are taught step by step.

**Dimensionality Reduction: Screening** We propose an additional technique for further dimensionality reduction when solving the reduced problem (8). The idea is to perform *screening*, which quickly identifies the zero components in $q^\star$ and removes them from the optimization problem. Leveraging DPC screening for non-negative lasso [16], we introduce a *screening* strategy for solving a general simplex-constrained linear least-squares problem (one can check problem (8) is indeed of this form):

$$\text{minimize} \quad \|X\beta - y\|_2^2, \qquad \text{subject to} \quad \beta \succeq 0, \|\beta\|_1 = 1. \tag{11}$$

We start with the following non-negative lasso problem, which is closely related to problem (11):

$$\text{minimize} \quad \phi_\lambda(\beta) := \|X\beta - y\|_2^2 + \lambda\|\beta\|_1, \qquad \text{subject to} \quad \beta \succeq 0, \tag{12}$$

and denote its solution by $\beta^\star(\lambda)$. One can show that if $\|\beta^\star(\lambda^\star)\|_1 = 1$, then $\beta^\star(\lambda^\star)$ is a solution to problem (11). Our screening strategy for problem (11) runs the DPC screening algorithm on the non-negative lasso problem (12), which applies a repeated screening rule (called EDPP) to solve a solution path specified by a $\lambda$-sequence: $\lambda_{max} = \lambda_0 > \lambda_1 > \cdots$. The $\ell_1$-norms along the solution path are non-decreasing: $0 = \|\beta^\star(\lambda_0)\|_1 \leq \|\beta^\star(\lambda_1)\|_1 \leq \cdots$. We terminate the solution path at $\lambda_t$ if $\|\beta^\star(\lambda_t)\|_1 \geq 1$ and $\|\beta^\star(\lambda_{t-1})\|_1 < 1$. Our goal is to use $\beta^\star(\lambda_t)$ to predict the zero components in $\beta^\star(\lambda^\star)$, a solution to problem (11). More specifically, we assume that the zero components in $\beta^\star(\lambda_t)$ are also zero in $\beta^\star(\lambda^\star)$, hence we can remove those components from $\beta$ (also the corresponding columns of $X$) in problem (11) and reduce its dimensionality.

While in practice this assumption is usually true provided that we have a delicate solution path, the monotonicity of $\beta^\star(\lambda)$'s support along the solution path does not hold in general [17]. Nevertheless, the assumption does hold when $\|\beta^\star(\lambda_t)\|_1 \to 1$, since the solution path is continuous and piecewise linear [18]. Therefore, we carefully design a solution path in the hope of a $\beta^\star(\lambda_t)$ whose $\ell_1$-norm is close to 1 (e.g. let $\lambda_i = \gamma\lambda_{i-1}$ with a large $\gamma \in (0,1)$, while more sophisticated design is possible such as a bi-section search). To remedy the (rare) situations where $\beta^\star(\lambda_t)$ predicts some incorrect zero components in $\beta^\star(\lambda^\star)$, one can always leverage the KKT conditions of problem (11) as a final check to correct those mis-predicted components [19]. Finally, note that the screening strategy may fail when the $\ell_1$-norms along the solution path converge to a value less than 1. In these cases we can never find a desired $\lambda_t$ with $\|\beta^\star(\lambda_t)\|_1 \geq 1$. In theory, such failure can be avoided by a modified lasso problem which in practice does not improve efficiency much (see the supplementary material).

## 4.2 The $w$-Solver: for Rule Selection

If we fix $p$, the optimization problem (3) boils down to:

$$\text{minimize} \quad \mathcal{E}(p,w;A,b) + \lambda_w P_w(w) \tag{13}$$
$$\text{subject to} \quad w \in \Delta^m.$$

We solve problem (13) via ADMM [20]:

$$w^{(k+1)} = \arg\min_w \ e^\top w + \lambda_w P_w(w) + \tfrac{\rho}{2}\|w - z^{(k)} + u^{(k)}\|_2^2, \tag{14}$$

$$z^{(k+1)} = \arg\min_z \ I_{\Delta^m}(z) + \tfrac{\rho}{2}\|w^{(k+1)} - z + u^{(k)}\|_2^2, \tag{15}$$

$$u^{(k+1)} = u^{(k)} + w^{(k+1)} - z^{(k+1)}. \tag{16}$$

In the $w$-update (14), we introduce the error vector $e = (Ap - b)^2$ (element-wise square), and obtain a closed-form solution by a soft-thresholding procedure [21]: for $r = 1, \ldots, K$,

$$w_{g_r}^{(k+1)} = \left(1 - \frac{\lambda_w \alpha \sqrt{m_r}}{(\rho + 2\lambda_w(1-\alpha)) \cdot \|\tilde{e}_{g_r}^{(k)}\|_2}\right)_+ \tilde{e}_{g_r}^{(k)}, \quad \text{where } \tilde{e}^{(k)} = \frac{\rho(z^{(k)} - u^{(k)}) - e}{\rho + 2\lambda_w(1-\alpha)}. \tag{17}$$

In the $z$-update (15), we introduce the indicator function $I_{\Delta^m}(z) = 0$ if $z \in \Delta^m$ and $\infty$ otherwise, and recognize it as a (Euclidean) projection onto the probability simplex:

$$z^{(k+1)} = \Pi_{\Delta^m}(w^{(k+1)} + u^{(k)}), \tag{18}$$

which can be solved efficiently by a non-iterative method [22]. Given that ADMM enjoys a linear convergence rate in general [23] and the problem's dimension $m \ll n$, one execution of the $w$-solver is cheaper than that of the $p$-solver. Indeed, the result from the $w$-solver can speed up the subsequent execution of the $p$-solver, since we can leverage the zero components in $w^\star$ to remove the corresponding rows in $A$, yielding additional savings in the group de-overlap of the $p$-solver.

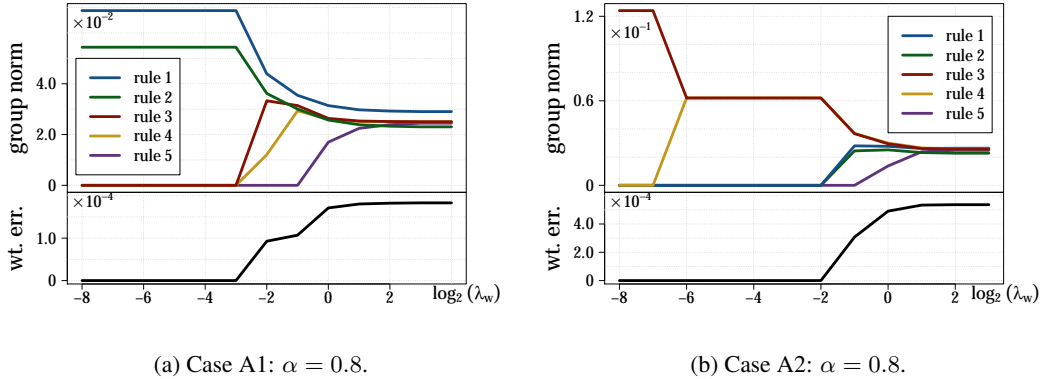

(a) Case A1: $\alpha = 0.8$.          (b) Case A2: $\alpha = 0.8$.

Figure 2: The $\lambda_w$-solution paths obtained from the two artificial rule sets. Each path is depicted by the trajectories of the group norms (top) and the trajectory of the weighted errors (bottom).

## 5 Experiments

### 5.1 Artificial Rule Set

We generate two artificial rule sets: Case A1 and A2, both of which are derived from the same raw input space $\mathcal{X} = \{x_1, \ldots, x_n\}$ for $n = 600$, and comprise $K = 5$ rules. The rules in Case A1 are of size $80, 50, 60, 60, 60$, respectively; the rules in Case A2 are of size $70, 50, 65, 65, 65$, respectively. For both cases, rule 1&2 and rule 3&4 are the only two consistent sub rule sets of size $\geq 2$. The main difference between the two cases is: in Case A1, rule 1&2 has a combined size of 130 which is larger than rule 3&4 and in Case A2 it is opposite. Under different settings of the hyperparameters $\lambda_w$ and $\alpha$, our model selects different rule combinations exhibiting unique "personal" styles.

Tuning the blending factor $\alpha \in [0, 1]$ is relatively easy, since it is bounded and has a nice interpretation. Intuitively, if $\alpha \to 0$, the effect of the group lasso vanishes, yielding a solution $w^\star$ that is not selective; if $\alpha \to 1$, the group elastic net penalty reduces to the group lasso, exposing the pitfall mentioned in Sec. 3. Experiments show that if we fix a small $\alpha$, the model picks either all five rules or none; if we fix a large $\alpha$, the group norms associated with each rule are highly unstable as $\lambda_w$ varies. Fortunately in practice, $\alpha$ has a wide middle range (typically between $0.4$ and $0.9$), within which all corresponding $\lambda_w$-solution paths look similar and perform stable rule selection. Therefore, for all experiments herein, we fix $\alpha = 0.8$ and study the behavior of the corresponding $\lambda_w$-solution path.

We show the $\lambda_w$-solution paths in Fig. 2. Along the path, we plot the group norms (top, one curve per rule) and the weighted errors (bottom). The former, formulated as $\|w^\star_{g_r}(\lambda_w)\|_2$, describes the options for rule selection; the latter, formulated as $\mathcal{E}(p^\star(\lambda_w), w^\star(\lambda_w); A, b)$, describes the quality of rule realization. To produce the trajectories, we start with a moderate $\lambda_w$ (e.g. $\lambda_w = 1$), and gradually increase and decrease its value to bi-directionally grow the curves. We terminate the descending direction when $w^\star(\lambda_w)$ is not group selective and terminate the ascending direction when the group norms converge. Both terminations are indicated by Thm. 1, and work well in practice. As $\lambda_w$ grows, the model transitions its compositional behavior from a conservative style (sacrifice a number of rules for accuracy) towards a more liberal one (sacrifice accuracy for more rules). If we further focus on the $\lambda_w$s that give us zero weighted error, Fig. 2a reveals rule 1&2, and Fig. 2b reveals rule 3&4, i.e. the largest consistent subset of the given rule set in both cases (Thm. 2). Finally, we mention the efficiency of our algorithm. Averaged over several runs on multiple artificial rule sets of the same size, the run-time of our solver is $27.2 \pm 5.5$ seconds, while that of a generic solver (CVX) is $41.4 \pm 3.8$ seconds. We attribute the savings to the dimensionality reduction techniques introduced in Sec. 4.1, which will be more significant at large scale.

### 5.2 Real Compositional Rule Set

As a real-world application, we test our unified framework on rule sets from an automatic music theorist [11]. The auto-theorist teaches people to write 4-part chorales by providing personalized

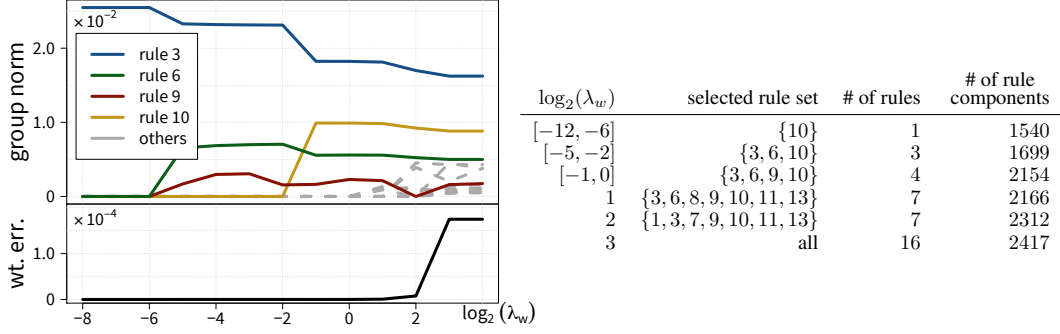

| $\log_2(\lambda_w)$ | selected rule set | # of rules | # of rule components |
|---|---|---|---|
| $[-12, -6]$ | $\{10\}$ | 1 | 1540 |
| $[-5, -2]$ | $\{3, 6, 10\}$ | 3 | 1699 |
| $[-1, 0]$ | $\{3, 6, 9, 10\}$ | 4 | 2154 |
| 1 | $\{3, 6, 8, 9, 10, 11, 13\}$ | 7 | 2166 |
| 2 | $\{1, 3, 7, 9, 10, 11, 13\}$ | 7 | 2312 |
| 3 | all | 16 | 2417 |

Figure 3: The $\lambda_w$-solution path obtained from a real compositional rule set.

rules at every stage of composition. In this experiment, we exported a set of 16 compositional rules which aims to guide a student in writing the next sonority that follows well with the existing music content. Each voice in a chorale is drawn from $\Omega = \{\texttt{R}, \texttt{G}_1, \ldots, \texttt{C}_6\}$ that includes the rest ($\texttt{R}$) and 54 pitches ($\texttt{G}_1$ to $\texttt{C}_6$) from human vocal range. The resulting raw input space $\mathcal{X} = \Omega^4$ consists of $n = 55^4 \approx 10^7$ sonorities, whose distribution lives in a very high dimensional simplex. This curse of dimensionality typically fails most of the generic solvers in obtaining an acceptable solution within a reasonable amount of time.

We show the $\lambda_w$-solution path associated with this rule set in Fig. 3. Again, the general trend shows the same pattern here: the model turns into a more liberal style (more rules but less accurate) as $\lambda_w$ increases. Along the solution path, we also observe that the consistent range (i.e. the error-free zone) is wider than that in the artificial cases. This is intuitive, since a real rule set should be largely consistent with minor contradictions, otherwise it will confuse the student and lose its pedagogical purpose. A more interesting phenomenon occurs when the model is about to leave the error-free zone. When $\log_2(\lambda_w)$ goes from 1 to 2, the combined size of the selected rules increases from 2166 to 2312 but the realization error increases only a little. Will sacrificing this tiny error be a smarter decision to make? The difference between the selected rules at these two moments shows that rule 1 and 7 were added into the selection at $\log_2(\lambda_w) = 2$ replacing rule 6 and 8. Rule 1 is about the bass line, while rule 6 is about tenor voice. It is known in music theory that outer voices (soprano and bass) are more characteristic and also more identifiable than inner voices (alto and tenor) which typically stay more or less stationary as background voices. So it is understandable that although larger variety in the bass increases the opportunity for inconsistency (in this case not too much), it is a more important rule to keep. Rule 7 is about the interval between soprano and tenor, while rule 8 describes a small feature between the upper two voices but does not have a meaning yet in music theory. So unlike rule 7 that brings up the important concept of voicing (i.e. classifying a sonority into open/closed/neutral position), rule 8 could simply be a miscellaneous artifact. To conclude, in this particular example, we would argue that the rule selection happens at $\log_2(\lambda_w) = 2$ is a better decision, in which case the model makes a good compromise on exact consistency.

To compare a selective rule realization with its non-selective counterpart [11], we plot the errors $\|A^{(r)}p - b^{(r)}\|_2$ for each rule $r = 1, \ldots, 16$ as histograms in Fig. 4. The non-selective realization takes all rules into consideration with equal importance, which turns out to be a degenerate case along our model's solution path for $\log_2(\lambda_w) \to \infty$. This realization yields a "well-balanced" solution but no rules are satisfied exactly. In constrast, a selective realization (e.g. $\log_2(\lambda_w) = 1$) gives near-zero errors on selected rules, producing more human-like compositional decisions.

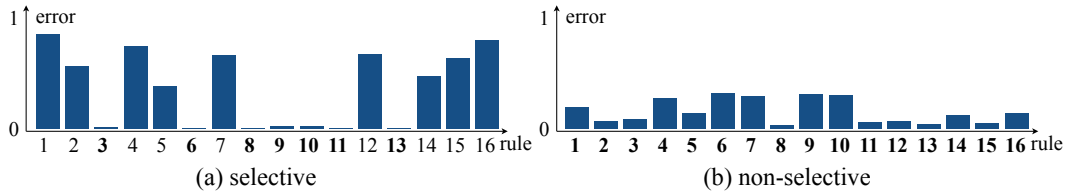

(a) selective        (b) non-selective

Figure 4: Comparison between a selective rule realization ($\log_2(\lambda_w) = 1$) and its non-selective counterpart. The boldfaced $x$-tick labels designate the indices of the selected rules.

# 6 Discussion

**Generality of the Framework**    The formalism of abstraction and realization in Sec. 2, as well as the unified framework for simultaneous rule realization and selection in Sec. 3, is general and domain-agnostic, not specific to music. The problem formulation as a bi-convex problem (3) admits numerous real-world applications that can be cast as (quasi-)linear systems, possibly equipped with some group structure. For instance, many problems in physical science involve estimating unknowns $x$ from their observations $y$ via a linear (or linearized) equation $y = Ax$ [24], where a grouping of $y_i$'s (say, from a single sensor or sensor type) itself summarizes $x$ as a rule/abstraction. In general, the observations are noisy and inconsistent due to errors from the measuring devices or even the failure of a sensor. It is then necessary to assign a different reliability weight to every individual sensor reading, and ask for a "selective" algorithm to "realize" the readings respecting the group structure. So in cases where some devices fail and give inconsistent readings, we can run the proposed algorithm to filter them out.

**Linearity versus Expressiveness**    The linearity with respect to $p$ in the rule system $Ap = b$ results directly from adopting the probability-space representation. However, this does not imply that the underlying domain (e.g. music) is as simple as linear. In fact, the abstraction process can be highly nonlinear which involves hierarchical partitioning of the input space [11]. So, instead of running the risk of losing expressiveness, the linear equation $Ap = b$ hides the model complexity in the $A$ matrix. On the other hand, the linearity with respect to $w$ in the bi-convex objective (3) is a design choice. We start with a simple linear model to represent relative importance for the sake of interpretability, which may sacrifice the model's expressiveness like other classic linear models. To push the boundary of this trade off in the future, we will pursue more expressive models without compromising (practically important) interpretability.

**Differences from (Group) Lasso**    Component-wise, both subproblems (7) and (13) of the unified framework look similar to regular feature selection settings such as lasso [25] and group lasso [26]. However, not only does the strong coupling between the two subproblems exhibit new properties (Thm. 1 and 2), but also the differences in the formulation present unique algorithmic challenges. First, the weighted error term (2) in the objective is in stark contrast with the regular regression formulation where (group) lasso is paired with least-squares or other similar loss functions. Whereas dropping features in a regression model typically increases training loss (under-fitting), dropping rules, on the contrary, helps drive the error to zero since a smaller rule set is more likely to achieve consensus. Hence, the tendency to drop rules in a regular (group) lasso is against the pursuit of a *largest* consistent rule set as desired. This stresses the necessity of a more carefully designed penalty like our proposed group elastic net. Second, the additional simplex constraint weakens the grouping property of group lasso: failures in group selection (i.e. there exists a rule that is not entirely selected) are observed for small $\lambda_w$s. The simplex constraint, effectively an $\ell_1$ constraint, also incurs an "$\ell_1$ cancellation", which nullifies a simple lasso (also an $\ell_1$) on a simple parameterization of the rules (one weight per rule). These differences pose new model behaviors and deserve further study.

**Local Convergence**    We solve the bi-convex problem (3) via alternating minimizations in which the algorithm decreases the non-negative objective in every iteration thus assures its convergence. Nevertheless, neither a global optimum nor a convergence in solution can be guaranteed. The former leaves the local convergence susceptible to different initializations, demanding further improvements through techniques such as random start and noisy updates. The latter leaves the possibility for the optimization variables to enter a limit cycle. However, we consider this as an advantage, especially in music where one prefers multiple realizations and interpretations that are equally optimal.

**More Microscopic Views**    The weighting scheme in this paper presents the rule selection problem in a most general setting, where a different weight is assigned to every rule component. Hence, we can study the relative importance not only *between* rules by the group norms $\|w_{g_r}\|_2$, but also *within* every single rule. The former compares compositional rules in a macroscopic level, e.g. restricting to a diatonic scale is more important than avoiding parallel octaves; while the latter in a microscopic level, e.g. changing the probability mass within a diatonic scale creates variety in modes: think about C major versus A minor. We can further study the rule system microscopically by sharing weights of the same component but from different rules, yielding an overlapping group elastic net.

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
