[Supplementary Material]

# Supplementary Material to "Probabilistic Rule Realization and Selection"

**Haizi Yu**[*][†]
Department of Computer Science
University of Illinois at Urbana-Champaign
Urbana, IL 61801
haiziyu7@illinois.edu

**Tianxi Li**[*]
Department of Statistics
University of Michigan
Ann Arbor, MI 48109
tianxili@umich.edu

**Lav R. Varshney**[†]
Department of Electrical and Computer Engineering
University of Illinois at Urbana-Champaign
Urbana, IL 61801
varshney@illinois.edu

## 1 Proofs of Main Theorems

We copy the bi-convex problem from the main paper:

$$\begin{aligned} \text{minimize} \quad & \mathcal{E}(p, w; A, b) + \lambda_p P_p(p) + \lambda_w P_w(w) \\ \text{subject to} \quad & p \in \Delta^n, w \in \Delta^m, \end{aligned} \tag{1}$$

where

$$\mathcal{E}(p, w; A, b) = (Ap - b)^\top \mathbf{diag}(w)(Ap - b),$$

$$P_p(p) = \sum_{r'=1}^{K'} \|p_{g'_{r'}}\|_1^2,$$

$$P_w(w) = \alpha \sum_{r=1}^{K} \sqrt{m_r} \|w_{g_r}\|_2 + (1 - \alpha)\|w\|_2^2.$$

Introduce $e = (Ap - b)^2$ where $(\cdot)^2$ is taken element-wise, then the weighted error is written as:

$$\mathcal{E}(p, w; A, b) = e^\top w = \sum_{r=1}^{K} e_{g_r}^\top w_{g_r}.$$

Recall that $w \in \Delta^m$ is *group selective* if for every rule in the rule set, $w$ either drops it or selects it entirely, i.e. either $w_{g_r} = 0$ or $w_{g_r} > 0$ element-wise, for all $r = 1, \ldots, K$.

Recall that for a group selective $w$, we introduce $\text{supp}_g(w)$ to be the set of the selected rule indices, i.e. $\text{supp}_g(w) = \{r \mid w_{g_r} > 0 \text{ element-wise}\} \subset \{1, \ldots, K\}$.

We start with a lemma which will be used in the proof of Thm. 1 part (1).

---

[*]Equal contribution.
[†]Supported in part by the IBM-Illinois Center for Cognitive Computing Systems Research (C3SR), a research collaboration as part of the IBM Cognitive Horizons Network.

**Lemma 1.** Fix any $p \in \mathbb{R}^n$, the bi-convex problem (1) is reduced to a convex problem w.r.t. $w$:

$$\text{minimize} \quad \sum_{r=1}^{K} e_{g_r}^\top w_{g_r} + \lambda_w (\alpha \sum_{r=1}^{K} \sqrt{m_r} \|w_{g_r}\|_2 + (1-\alpha)\|w\|_2^2) \tag{2}$$

$$\text{subject to} \quad \mathbf{0} \preceq w \preceq \mathbf{1}, \;\; \mathbf{1}^\top w = 1.$$

Let $w^\star$ be a solution to problem (2), then $w^\star$ is group selective if

$$\lambda_w > \alpha^{-1}\|e\|_\infty.$$

*Proof of Lemma 1.* (Proof by contradiction) Assume $w^\star$ is not group selective when $\lambda_m > \alpha^{-1}\|e\|_\infty$, i.e. there exists $r \in \{1, \dots, K\}$ such that

$$\|w_{g_r}^\star\|_2 \neq 0, \text{ but } (w_{g_r}^\star)_i = 0, \text{ for some } i.$$

Let $L(w)$ be problem (2)'s objective function whose partial derivative $\partial L/\partial (w_{g_k})_j$ for any group $k$ such that $\|w_{g_k}^\star\|_2 \neq 0$ is

$$\frac{\partial L}{\partial(w_{g_k})_j}(w^\star) = (e_{g_k})_j + \lambda_w \left( \alpha\sqrt{m_r} \frac{(w_{g_k}^\star)_j}{\|w_{g_k}^\star\|_2} + 2(1-\alpha)(w_{g_k}^\star)_j \right).$$

For $r, i$ in particular, we have

$$\frac{\partial L}{\partial(w_{g_r})_i}(w^\star) = (e_{g_r})_i.$$

Within group $r$, let $j^\star = \arg\max_j (w_{g_r}^\star)_j$. Since $\sum_{j=1}^{m_r}(w_{g_r}^\star)_j^2 = \|w_{g_r}^\star\|_2^2 > 0$, we must have

$$(w_{g_r}^\star)_{j^\star} \geq \frac{1}{\sqrt{m_r}}\|w_{g_r}^\star\|_2. \tag{3}$$

Take the direction vector $\delta^t \in \mathbb{R}^m$ such that

$$(\delta_{g_k}^t)_j = \begin{cases} t & \text{if } k=r, j=i \\ -t & \text{if } k=r, j=j^\star \\ 0 & \text{otherwise}, \end{cases}$$

where $t > 0$ is sufficiently small such that $w^\star + \delta^t \in \Delta^m$. However,

$$\left\langle \frac{\partial L}{\partial w^\star}, \delta^t \right\rangle = t \cdot (e_{g_r})_i - t(e_{g_r})_{j^\star} - t\lambda_w \left( \alpha\sqrt{m_r} \frac{(w_{g_r}^\star)_{j^\star}}{\|w_{g_r}^\star\|_2} + 2(1-\alpha)(w_{g_r}^\star)_{j^\star} \right)$$

$$\leq t \cdot (e_{g_r})_i - t\lambda_w \alpha \sqrt{m_r} \frac{(w_{g_r}^\star)_{j^\star}}{\|w_{g_r}^\star\|_2} \tag{4}$$

$$\leq t \left( (e_{g_r})_i - \lambda_w \alpha \right) \tag{5}$$

$$< 0 \tag{6}$$

where inequality (4) holds since $(e_{g_r})_{j^\star} \geq 0, \alpha \leq 1, (w_{g_k}^\star)_{j^\star} \geq 0$; inequality (5) holds because of the lower bound (3); inequality (6) holds because of the condition $\lambda_w > \alpha^{-1}\|e\|_\infty$. The negative inner product in the above implies that $\delta^t$ is a descent direction, which gives

$$L(w^\star + \delta^t) < L(w^\star).$$

This contradicts the fact that $w^\star$ is a minimizer, therefore, nullifies the assumption that $w^\star$ is not group selective and completes the proof.

$\square$

**Theorem 1.** *Fix any $\lambda_p > 0, \alpha \in [0,1]$. Let $(p^\star(\lambda_w), w^\star(\lambda_w))$ be a solution path to problem (1).*
*(1) $w^\star(\lambda_w)$ is group selective, if $\lambda_w > 1/\alpha$.*
*(2) $\|w_{g_r}^\star(\lambda_w)\|_2 \to \sqrt{m_r}/m$ as $\lambda_w \to \infty$, for $r = 1, \dots, K$.*

*Proof of Theorem 1.* For part (1), recall that $A^{(r)}p, b^{(r)} \in \Delta^{m_r}$ are both probability distributions regarding the $r$th rule. Hence, every element of $e_{g_r} = (A^{(r)}p - b^{(r)})^2$ is bounded in $[-1, 1]$, for any $r = 1, \ldots, K$. That is, $e_i \in [-1, 1]$ for all $i$, or equivalently $\|e\|_\infty \leq 1$. Notice that $w^\star(\lambda_w)$ is the solution to problem (2) for $p^\star(\lambda_w)$. Then by Lemma 1, $w^\star(\lambda_w)$ is group selective since $\lambda_w > 1/\alpha \geq \|e\|_\infty/\alpha$.

For part (2), when $\lambda_w \to \infty$, the first two terms in the objective vanish, problem (1) is equivalent to

$$\text{minimize} \quad \alpha \sum_{r=1}^{K} \sqrt{m_r} \|w_{g_r}\|_2 + (1 - \alpha)\|w\|_2^2 \tag{7}$$

$$\text{subject to} \quad \mathbf{0} \preceq w \preceq \mathbf{1}, \quad \mathbf{1}^\top w = 1.$$

Notice that the solution $w^\star$ to problem (7) must have uniform mass within each group, i.e.

$$(w_{g_r}^\star)_i = \|w_{g_r}^\star\|_1/m_r, \quad \text{for all } i = 1, \ldots, m_r, \text{ and for all } r = 1, \ldots, K. \tag{8}$$

As a consequence, the group lasso penalty is always constant:

$$\sum_{r=1}^{K} \sqrt{m_r} \|w_{g_r}^\star\|_2 = \sum_{r=1}^{K} \sqrt{m_r} \frac{\|w_{g_r}^\star\|_1}{\sqrt{m_r}} = \|w^\star\|_1 = 1.$$

Under the uniformity condition (8), the ridge penalty

$$\|w\|_2^2 = \sum_r \|w_{g_r}\|_2^2 = \sum_r \frac{\|w_{g_r}\|_1^2}{m_r} \geq \frac{(\sum_r \|w_{g_r}\|_1)^2}{\sum_r m_r} = \frac{1}{m},$$

by the Cauchy-Schwarz inequality. The equality holds when

$$\frac{\|w_{g_r}^\star\|_1}{\sqrt{m_r}} = \gamma\sqrt{m_r}, \quad \text{for some constant } \gamma.$$

That is $\|w_{g_r}^\star\|_1 = \gamma m_r$. Then $\sum_r \|w_{g_r}^\star\|_1 = 1$ gives $\gamma = 1/m$, which further yields

$$\|w_{g_r}^\star\|_2 = \frac{\|w_{g_r}^\star\|_1}{\sqrt{m_r}} = \gamma\sqrt{m_r} = \frac{\sqrt{m_r}}{m}.$$

This completes the proof and further shows that $w^\star$ is actually the uniform distribution on $\Delta^m$, since applying the uniformity condition (8) to $\|w_{g_r}^\star\|_1 = \gamma m_r = m_r/m$ gives $w_i^\star = 1/m$ for all $i$.

$\square$

**Theorem 2.** *For $\lambda_p = 0$ and any $\lambda_w > 0, \alpha \in [0, 1]$, let $(p^\star, w^\star)$ be a solution to problem* (1). *We define $\mathcal{C} \subset 2^{\{1,\ldots,K\}}$ such that any $C \in \mathcal{C}$ is a consistent (error-free) subset of the given rule set. If $\text{supp}_g(w^\star) \in \mathcal{C}$, then $\sum_{r \in \text{supp}_g(w^\star)} m_r = \max\left\{\sum_{r \in C} m_r \mid C \in \mathcal{C}\right\}$.*

*Proof of Theorem 2.* When $\lambda_p = 0$, problem (1) becomes

$$\text{minimize} \quad \mathcal{E}(p, w; A, b) + \lambda_w\left(\alpha \sum_{r=1}^{K} \sqrt{m_r} \|w_{g_r}\|_2 + (1 - \alpha)\|w\|_2^2\right) \tag{9}$$

$$\text{subject to} \quad \mathbf{0} \preceq p \preceq \mathbf{1}, \quad \mathbf{1}^\top p = 1,$$
$$\mathbf{0} \preceq w \preceq \mathbf{1}, \quad \mathbf{1}^\top w = 1.$$

Under the error-free condition, it is clear that $(p^\star, w^\star)$ is also the solution to the following problem

$$\text{minimize} \quad \lambda_w \alpha \sum_{r=1}^{K} \sqrt{m_r} \|w_{g_r}\|_2 + \lambda_w(1 - \alpha)\|w\|_2^2 \tag{10}$$

$$\text{subject to} \quad \text{supp}_g(w) \in \mathcal{C}$$
$$\mathcal{E}(p, w; A, b) = 0$$
$$\mathbf{0} \preceq p \preceq \mathbf{1}, \quad \mathbf{1}^\top p = 1.$$
$$\mathbf{0} \preceq w \preceq \mathbf{1}, \quad \mathbf{1}^\top w = 1.$$

which can be further rewritten as the following problem by the definition of $\mathcal{C}$:

$$\text{minimize} \quad \lambda_w \alpha \sum_{r=1}^{K} \sqrt{m_r} \|w_{g_r}\|_2 + \lambda_w (1 - \alpha) \|w\|_2^2 \tag{11}$$
$$\text{subject to} \quad \text{supp}_g(w) \in \mathcal{C}$$
$$\mathbf{0} \preceq w \preceq \mathbf{1}, \quad \mathbf{1}^\top w = 1.$$

The above problem is further equivalent to the following problem

$$\text{minimize}_{C,w} \quad \lambda_w \alpha \sum_{r=1}^{K} \sqrt{m_r} \|w_{g_r}\|_2 + \lambda_w (1 - \alpha) \|w\|_2^2 \tag{12}$$
$$\text{subject to} \quad \text{supp}_g(w) = C, \ C \in \mathcal{C},$$
$$\mathbf{0} \preceq w \preceq \mathbf{1}, \quad \mathbf{1}^\top w = 1.$$

Consider a class of problems $\{Q(C) \mid C \in \mathcal{C}\}$ such that each problem $Q(C)$ is formulated as

$$\text{minimize}_w \quad \lambda_w \alpha \sum_{r=1}^{K} \sqrt{m_r} \|w_{g_r}\|_2 + \lambda_w (1 - \alpha) \|w\|_2^2 \tag{13}$$
$$\text{subject to} \quad \text{supp}_g(w) = C$$
$$\mathbf{0} \preceq w \preceq \mathbf{1}, \quad \mathbf{1}^\top w = 1.$$

The solution $w^\star$ to problem (12) is then the solution to problem (13) with the minimum objective among all the problems from $\{Q(C) \mid C \in \mathcal{C}\}$, and further the corresponding $p^\star$ is probability distribution that gives $\mathcal{E}(p^\star, w^\star; A, b) = 0$.

Given any $C \in \mathcal{C}$, let $(p^{\star,C}, w^{\star,C})$ be the solution to the corresponding problem (13). This problem is a reduced problem (7) introduced in the proof of Thm. 1. The only difference is that here we constrain the nonzero groups to be $C$. Same argument for problem (7) gives

$$w_{g_r}^{\star,C} = \frac{1}{\sum_{r \in C} m_r} \mathbf{1}_{m_r}, \quad \text{for any } r \in C.$$

Thus, the optimal objective of $Q(C)$ for a given $C$ is

$$\lambda_w \alpha + \lambda_w (1 - \alpha) \frac{1}{\sum_{r \in C} m_r}.$$

Finally, as $w^\star$ is the $w^{\star,C}$ that achieves the minimum $Q(C)$ objective among all $C \in \mathcal{C}$, we have

$$\sum_{r \in \text{supp}_g(w^\star)} m_r = \max \left\{ \sum_{r \in C} m_r \mid C \in \mathcal{C} \right\}.$$

$\square$

## 2 Theoretical Justification for Screening

The screening procedure is needed when we solve for $p$ after fixing $w$. In this section, we give detailed justifications about the procedure in its general form: a simplex constrained least-squares:

$$\text{minimize} \quad \frac{1}{2} \|Ap - b\|_2^2 \tag{14}$$
$$\text{subject to} \quad \mathbf{0} \preceq p \preceq \mathbf{1}, \quad \mathbf{1}^\top p = 1,$$

We start from the observation that the solution $p^\star$ to the problem is expected to be sparse in most situations. To see this, we first introduce an relaxation problem for later discussions. Note that we

can relax (14) to the following $L_1$-constrained problem:

$$\text{minimize} \quad \frac{1}{2}\|Ap - b\|_2^2 \tag{15}$$
$$\text{subject to} \quad p \succeq 0$$
$$\|p\|_1 \leq 1$$

which is further is equivalent to the following problem:

$$\text{minimize} \quad \frac{1}{2}\|Ap - b\|_2^2 + \rho\|p\|_1 \tag{16}$$
$$\text{subject to} \quad p \succeq 0$$

for some positive (unknown) $\rho = \rho^\star$.

Denote the solution of (16) with $\rho$ to be $\hat{p}(\rho)$. We have the following result:

**Proposition 1.** Any solution of (16) satisfying

$$\|\hat{p}(\rho)\|_1 = 1. \tag{17}$$

is a solution of (14).

*Proof of Proposition 1.* See the proof of Proposition 2. □

It is well known that the lasso problem tends to encourage a sparse solution, so we may expect a sparse solution for $p^\star$ in (14). In practice, we observe that the solution of (14) is indeed very sparse in most experiments. This motivates us to screen out the positions of $p$ that must be zeros according to the nonnegative lasso solution before solving the least square problem. Since negative lasso screening and solving can be done efficiently [1] in many situations, such strategy can sometimes reduce the problem scale dramatically.

The difference between (16) and (14) comes from the fact that we are relaxing the equality of $L_1$ norm $\|p\|_1 = 1$ by the inequality $\|p\|_1 \leq 1$. Actually, in general they are not guaranteed to be equivalent. There are situations where the lasso solution path can never touch the $L_1$ unit ball and in such circumstances, the lasso solution cannot be used. However, we can make a small modification on (16) to achieve an exact equivalence between the simplex constrained least square problem and a nonnegative lasso problem. Specifically, we attach a row, $c\mathbf{1}^T$ to $A$ and $2c$ to $b$, and denote the resulting matrices by $\tilde{A}$ and $\tilde{b}$, where $c$ is some constant that will be given soon. The modified nonnegative lasso problem is

$$\text{minimize} \quad \frac{1}{2}\|\tilde{A}p - \tilde{b}\|_2^2 + \rho\|p\|_1 \tag{18}$$
$$\text{subject to} \quad p \succeq 0$$

The following proposition shows that the (18) is equivalent to (14).

**Proposition 2.** Let $T = \min_j \|A_{\cdot j}\|_2$. If $c^2 > 4T^2 + m$, then

1. There exists a $\rho$, such that the solution of problem (18), $\hat{p}(\rho)$, satisfies (17).

2. In addition, such $\hat{p}(\rho)$ must be a solution of problem (14).

*Proof of Proposition 2.* We first check the least square problem

$$\text{minimize} \quad \frac{1}{2}\|\tilde{A}p - \tilde{b}\|_2^2 \tag{19}$$
$$\text{subject to} \quad p \succeq 0$$

For any $p$ such that $\|p\|_1 < 1$, the last term in the squared error:

$$(c\mathbf{1}^T p - 2c)^2 > c^2.$$

So $\|\tilde{A}p - \tilde{b}\|_2^2 > c^2$. On the other hand, let $j^* = \arg\min_j \|A_{.j}\|_2$, and set $\tilde{p}$ to be the vector with all zeros except at $j^*$ coordinate that is 2 so $\|\tilde{p}\|_1 = 2$. Then

$$\|\tilde{A}\tilde{p} - \tilde{b}\|_2^2 = \|2A_{.j} - b\|^2 \le 4\|A_{.j}\|_2 + \|b\|_2^2 \le 4T^2 + m < c^2,$$

since $b_i \le 1$ for any $i$. That means, for any $\|p\| < 1$, $\tilde{p}$ has a smaller objective, thus the solution of the problem (19) must not lie in the interior of the unit $L_1$ ball. Define the problem

$$\begin{aligned} \text{minimize} \quad & \frac{1}{2}\|Ap - b\|_2^2 \\ \text{subject to} \quad & p \succeq 0 \\ & \|p\|_1 \le 1 \end{aligned} \tag{20}$$

Let the solution of (20) be $p^\star$ and the solution of (19) be $p^*$. We have shown $\|p^*\|_1 \ge 1$. Now we prove $\|p^\star\|_1 = 1$ by contradiction. Assume $\|p^\star\|_1 < 1$ so we know $p^\star \neq p^*$. Then due to the continuity of $L_1$ norm, there must exist $t \in (0, 1)$ such that the convex combination of $p^\star$ and $p^*$: $p_t = tp^\star + (1 - t)p^*$ satisfies

$$\|p_t\|_1 = 1.$$

Due to the strict convexity, we have

$$\|\tilde{A}p_t - \tilde{b}\|_2^2 \le t\|\tilde{A}p^\star - \tilde{b}\|_2^2 + (1 - t)\|\tilde{A}p^* - \tilde{b}\|_2^2 < \|\tilde{A}p^\star - \tilde{b}\|_2^2$$

This contradicts the definition of $p^\star$. Thereafter, the solution of problem (20) must be on the unit $L_1$ ball. Finally, (18) has a solution to satisfy (17) for some $\rho$ due to its equivalence with (20).

For the second part, given any solution $\hat{p}(\rho)$ of (18) satisfying (17), it is a feasible point of (14). The last term of the modified least square problem makes fixed contribution in the objective of (14) and does not change within the feasible region. Moreover, for any solution $p^\star$ of (14), assume

$$\|Ap^\star - b\|_2^2 < \|A\hat{p}(\rho) - b\|_2^2,$$

then we must have

$$\frac{1}{2}\|Ap^\star - b\|_2^2 + \rho\|p^\star\|_1 < \frac{1}{2}\|A\hat{p}(\rho) - b\|_2^2 + \rho\|\hat{p}(\rho)\|_1.$$

This contradicts the definition of $\hat{p}(\rho)$, thus we have

$$\|Ap^\star - b\|_2^2 = \|A\hat{p}(\rho) - b\|_2^2$$

and $\hat{p}(\rho)$ is a solution of (14). $\qquad\qquad\square$

In practice, we prefer to use Proposition 1 (or correspondingly, problem (16)) as the basis for screening whenever it is possible, even though Proposition 2 (or correspondingly, problem (18)) is always guaranteed to success in theory. This is because the modified problem (18) has very imbalanced design as the last row is much larger than the others. This often makes the screening less effective in the sense that many zeros positions cannot be detected. It will be interesting to investigate if there is alternative screening strategy that always works in theory with better practical efficiency. We leave this for future work.

## References

[1] J. Wang and J. Ye, "Two-layer feature reduction for sparse-group lasso via decomposition of convex sets," in *Proc. 28th Annu. Conf. Neural Inf. Process. Syst. (NIPS)*, 2014, pp. 2132–2140.