[Reviews · NeurIPS 2017]

Reviewer 1



This paper describes a method for selecting and weighting "rules" (equivalence class partitions of a discrete/binary feature space), and its application to music composition for 4-part chorales. Given a set of candidate rules, the proposed method learns a selection weighting to determine which rules to include, and a weighting over the selected rules to maximize consistency. Both problems are modeled as least-squares regression over the probability simplex (with differing group-sparsity penalties for the different variables), and the authors provide some efficiency improvements to keep the problem tractable for their application. Overall, the paper is well-written, and the design choices are well-motivated. That said, it's not obvious that NIPS is an appropriate venue for this work, for a couple of reasons: 1. There is only one application considered here, which is fairly niche, and maybe not of broad interest to this community. 2. The experimental evaluation focuses entirely on the properties of the solution vectors (e.g., solution paths as a function of regularization factors), but there is no empirical validation on the motivating application, so it's hard to view this as an "applications paper". 3. The algorithmic techniques used here are all standard and well-known (though their specific combination might be novel). The various solver optimizations (group de-overlap, column reduction) might be of broader interest, but it's not clear that they have broader applications than considered here. It doesn't seem likely that NIPS readers will take much away on the algorithm design front.

Reviewer 2



The paper tackles rules realisation and selection, given a specific set of given rules that derives from an abstraction process, in the specific case of music compositional rules. The novelty of the paper resides in a new bi-convex optimisation model for rules selection and rule realisation. The challenge posed by rules realisation is due to the high dimensionality of the sample space. The challenge posed by rule selection appears in the case when all rules cannot be simultaneously realised, or not in their entirety. In this case one has to choose which rules to respect and in which amount of components. The strengths of the paper are: - the goal and challenge of the model are very clearly stated, - the paper is extremely well written with all notations defined and intuitive explanations of the theoretical results. I especially liked the interpretation of lines 285 - 296. - experimental results support the claim of the proposed innovations, as the model (i) behaves as expected when varying the hyper parameters (ii) reduces computation time. Questions / comments : - line 118 : can the author explain why the solution is expected to be "unique" on p thanks to group penalty ? If there is no group penalty (\lambda_p = 0) I would expect the solution to be unique, which is also what the authors state in line 131 "in setting \lambda_p sufficiently small, we secure a unique solution for the rule realisation part". The two statements seem unconsistent. - line 148 : Theorem 1, claim (2) : am I right to understand that all rules are selected because for all rules the L2 norm converges to a non-zero value ? - Group De-overlap : am I right to understand that this is not an additional dimensionality reduction but only an interpretation of the grouping presented in Section (3), while screening is an additional step to further reduce dimensionality ? - line 304 : what do the authors mean by "failures in group selection" ? Does it mean that no rule is entirely selected ? - line 305 : am I right to understand that a simple lasso on the number of rules is impossible because it makes no sense to weight the rules as the selection process is a 0/1 process ? - I am not familiar with the literature in automatic music composition, and I understand that there is no "ground-truth" solution as creativity can be a desired behaviour. I like the fact that the author mention the gain in computational efficiency. If that exists / is possible in the time of rebuttal I would like to see a baseline model to compare the behaviour in terms of selection / realisation of the proposed model. - Can the author think of any other possible domain to apply their method ?

Reviewer 3



The authors present a solution to the problem of jointly selecting and realizing rules which they formulate as a biconvex optimization problem. The optimization exploits group structures in the discrete input space p and rule selection weights w. A group ridge penalty on the L1 norm is used on p to encourage diversity and a group elastic net is used on w with a hyperparameter alpha that controls the effect of the L1 and L2 penalties. Since the objective is biconvex (i.e) freezing p makes the objective convex in w and vice-versa, solvers for p and w are fairly straightforward. The authors also perform further dimensionality reduction by identifying and removing zero components in q during the optimization process. The p-solver exploits dimensionality reduction as a result of group structures. The rule selection (w-problem) is solved using ADMM. Experiments are carried out using an artificial rule set and data from [11]. Although paper presents a well motivated and scalable solution to the problem, the model parameterization is somewhat simple and uses fairly straightforward extensions of existing optimization techniques to learn the model parameters. The introduction is well written and the need for having interpretable realization models is well motivated. The simple linear parameterization of the model, while being able to offer interpretability, limits its expressiveness and applicability to more complicated application domains. I got the impression that the ideas expressed in Section 3 (until the Model Interpretation subsection) were well presented but at the cost of being a little too verbose, specifically the intuitions behind the hyperparameters lambda_p, lambda_w and alpha. I didn’t quite understand the motivation for the group elastic net formulation for rule selection (w), specifically about how it provides “more control over the number of selections”. Would it be interesting to consider sparsity within w groups using something like a sparse group elastic net? - https://arxiv.org/abs/1001.0736. While this is largely a theoretical paper, I found the experiments (especially 5.2) somewhat lacking more discussion. I would have liked to see some more qualitative analysis about realization process since a major motivation for this paper is interpretability. In conclusion, I think that while the paper provides and clean, scalable and easy to understand solution to an important problem, it is somewhat lacking in novelty in that it uses a fairly straightforward model parameterization and optimization solution without strong experimental backing.